# Mental Health and Quality of Life for Disaster Service Workers in a Province under COVID-19

**DOI:** 10.3390/jcm11061600

**Published:** 2022-03-14

**Authors:** Ji-Won Na, Chan-Mo Yang, Sang-Yeol Lee, Seung-Ho Jang

**Affiliations:** Department of Psychiatry, School of Medicine, Wonkwang University, Iksan 54538, Korea; skwldnjs9874123@naver.com (J.-W.N.); ikarosforeve@naver.com (C.-M.Y.); psysangyeol@hanmail.net (S.-Y.L.)

**Keywords:** COVID-19, disaster service workers, mental health, quality of life, resilience

## Abstract

Objective: Healthcare workers and disaster service workers have been reported to be vulnerable to mental health problems during outbreaks of infectious diseases such as the COVID-19 pandemic. This study aimed to investigate the psychosocial characteristics of disaster service workers in charge of COVID-19-related work and also identify the factors affecting their quality of life. Methods: From June 2020 to June 2021, a survey was conducted of 526 disaster service workers in charge of COVID-19-related work. This included those working in public health care centers (PHC), 119 rescue and emergency medical services (119 REMS), public servants of city hall (PS), and police officers. The Korean version of the Fear of COVID-19 Scale, Patient Health Questionnaire-15, Hospital Anxiety and Depression Scale, Insomnia Severity Index, Connor–Davidson Resilience Scale, and World Health Organization quality of life assessment instrument brief form were used. A one-way ANOVA was conducted, and a stepwise regression analysis was carried out to determine the factors affecting quality of life. Results: Regarding quality of life, 119 REMS (180.64 ± 26.20) scored significantly higher than PHC (165.76 ± 23.73) and PS (163.90 ± 23.60), while police officers (176.87 ± 23.17) scored significantly higher than PS (163.90 ± 23.60) (*F* = 12.373, *p* < 0.001). Resilience (β = 0.897, *p* < 0.01) was the most significant explanatory variable, and together with insomnia (β = 0.154, *p* < 0.01), depression (β = −0.152, *p* < 0.01), and COVID-19 anxiety (β = −0.057, *p* < 0.01) accounted for 91.8% of the explanatory variance with regard to quality of life. Discussion: Quality of life was found to be negatively correlated with insomnia, depression, and COVID-19 anxiety while being positively correlated with resilience. Therefore, active interventions are needed to improve the resilience of disaster service workers.

## 1. Introduction

On 12 January 2020, reported cases of pneumonia caused by a novel coronavirus “2019-nCov(COVID-19)” were designated as the coronavirus disease 2019 (COVID-19) by the World Health Organization (WHO). This disease spread rapidly [1]. On 30 January 2020, the WHO declared the outbreak of SARS-CoV-2 infections to be a global public health emergency of international concern [2]. As COVID-19 continued to spread, about 190 million cases were confirmed worldwide, and about 180,000 cases were confirmed in South Korea as of June 2021. The number of confirmed cases is rising rapidly with the emergence of highly contagious variants [3], and as South Korea has experienced repeated waves of the COVID-19 pandemic, more than 40 million out of 52 million people in South Korea have been vaccinated against COVID-19 [4]. Due to the pandemic, the governments of every country have implemented measures regarding hand hygiene, social distancing, and the use of personal protective equipment to prevent the spread of the virus [5], and South Korea is responding to the pandemic by educating the public on preventative measures, and using diagnostic kits to conduct rapid tests [6].

The clinical symptoms of COVID-19 include cough, fever, fatigue, and muscle aches, and in about 10% of patients, gastrointestinal symptoms such as nausea, diarrhea, and abdominal pain may precede other symptoms. In severe cases, infection can even lead to serious breathing difficulties [7]. The COVID-19 pandemic has not only caused physical illnesses but also led to global socioeconomic problems [8], and the biopsychosocial impact of COVID-19 is leading to serious mental health problems [9].

Infectious diseases are reported to have a psychosocial impact on society [10] and can have symptoms such as anxiety, fear, anger, stress, terror, insomnia, pain, boredom, loneliness, depression, and post-traumatic stress disorder (PTSD) in the public [11,12]. During the severe acute respiratory syndrome (SARS) outbreak that began in 2002, high levels of depression and anxiety were reported not only in patients but also in the public [13,14]. In addition, the Middle East respiratory syndrome (MERS) outbreak in 2015 was reported to cause a high prevalence of PTSD and depression, even 12 months after the outbreak [15].

In recent years, the COVID-19 pandemic has been one of the most stressful events worldwide [16]. When diagnosed with COVID-19, people experience difficulties such as feelings of isolation as a result of being quarantined, feelings of guilt toward their family, exclusion by the people around them, as well as job losses and subsequent financial problems [17]. The fear of exposure to the virus has also caused the public to experience high levels of anxiety and fear regarding the risk of infection and death [18]. In addition to the people who are infected or are self-isolating, those on the frontlines such as healthcare workers and disaster service workers have been reported to be vulnerable to mental health problems during outbreaks of infectious diseases such as the COVID-19 pandemic. Therefore, active interventions are necessary to promote the mental health of these workers [19]. However, there are few studies on the mental health of healthcare workers in relation to COVID-19 and hardly any studies evaluating that of disaster service workers. Therefore, this study aimed to investigate the psychosocial characteristics of disaster service workers in charge of COVID-19-related work, and also identify the factors affecting their quality of life.

## 2. Methods

### 2.1. Study Subjects

A survey was conducted of 526 disaster service workers in charge of COVID-19-related work who understood this study’s purpose and agreed to participate from June 2020 to June 2021, when mental health issues related to COVID-19 began to emerge. This included those working in public health care centers (PHC), 119 rescue and emergency medical services (119 REMS), public servants of city hall (PS), and police officers. The study was approved by the Wonkwang University Hospital Institutional Review Board (IRB) and received the written consent of all participants (IRB approval number: WKUH 2020-05-052). 

### 2.2. Measurement Tools 

#### 2.2.1. COVID-19 Anxiety 

To measure COVID-19 anxiety, this study used the KF-COVID-19S, which consists of seven items that are evaluated on a 5-point Likert scale. Jang et al. [20] developed this scale by translating the original version of the Fear of COVID-19 Scale (F-COVID-19S) into Korean and verified its reliability and validity. The total score ranges from 5 to 35 points with a cut-off score of 26 points.

#### 2.2.2. Somatic Symptoms 

To measure somatic symptoms, this study used the Patient Health Questionnaire-15 (PHQ-15), a self-report questionnaire developed by Kroenke et al. [21]. The PHQ-15 consists of 15 items that were evaluated on a 3-point Likert scale ranging from 0 to 2 points. The severity of somatic symptoms was determined by categorizing the total PHQ-15 scores into scores of 5 points or less, 6 to 10 points, and 11 or more points to represent low, medium, and high levels of somatic symptom severity, respectively. Han et al. [22] translated the questionnaire into Korean and verified its validity and reliability. 

#### 2.2.3. Anxiety and Depression

To measure anxiety and depression, this study used the Hospital Anxiety and Depression Scale (HADS) developed by Zigmond et al. [23]. In the HADS, the depression subscale (HAD-D) consists of seven even-numbered items and the anxiety subscale (HAD-A) consists of seven odd-numbered items comprising 14 items in total, which are evaluated on a 4-point Likert scale ranging from 0 to 3 points. The cut-off score for both the HAD-A and HAD-D is 8 points [23]. Oh et al. [24] translated the scale into Korean and verified its validity and reliability.

#### 2.2.4. Insomnia

To measure insomnia, this study used the Insomnia Severity Index (ISI) developed by Bastien et al. [25]. The ISI consists of seven items that assess the severity of the insomnia problems experienced in the last two weeks with regard to sleep satisfaction, concerns about sleep, quality of life impairments, and daytime functioning impairments. Each item is evaluated on a 5-point Likert scale, and those with a total score of 15 points or more are considered to be in a risk group for insomnia [25]. Cho et al. [26] translated the index into Korean and verified its validity and reliability. 

#### 2.2.5. Quality of Life

To measure the quality of life, this study used the WHOQOL-BREF, an abbreviated quality of life questionnaire developed by the World Health Organization Quality of Life Group [27]. It comprises 26 items including the following: two items on the overall quality of life, six items on the psychological domain, seven items on the physical domain, eight items on the environmental domain, and three items on the social relationships domain. Each item is evaluated on a scale ranging from 0 to 5 points with a score of 60 points as the optimal cut-off score and higher scores indicating a higher quality of life [28]. Min et al. [29] translated the questionnaire into Korean and verified its validity and reliability. 

#### 2.2.6. Resilience

To measure resilience, this study used the Connor–Davidson Resilience Scale (CD-RISC) developed by Conner and Davidson [30]. The CD-RISC consists of 25 items regarding five aspects of resilience, including hardiness, persistence, optimism, support, and spirituality. Each item is evaluated on a scale ranging from 0 to 4 points for a total score of 100 points, and higher scores indicate higher levels of resilience [30]. Baek et al. [31] translated the scale into Korean and verified its validity and reliability. 

### 2.3. Statistical Methods

This study compared the demographics and psychosocial characteristics of each group. The categorical variables showed the ratio and frequency, while the continuous variables showed the mean and standard deviation. A one-way ANOVA was conducted to compare the differences between three groups or more, and a Pearson correlation test was conducted to analyze the correlations between the quality of life by occupation and psychosocial characteristics. In addition, a stepwise regression analysis was conducted to determine the factors affecting the quality of life. Data were analyzed using the Statistical Package for the Social Sciences (SPSS, version 21; Chicago, IL, USA). 

## 3. Results

### 3.1. Demographic Characteristics of Study Subjects 

The most common demographic characteristics included the following: female (*n* = 267, 50.8%), 30s age group (*n* = 159, 30.2%), married (*n* = 303, 57.6%), and a college graduate education level or higher (*n* = 388, 73.8%). Occupations included PHC (*n* = 66, 12.5%), PS (*n* = 250, 47.5%), 119 REMS (*n* = 122, 23.2%), and police officer (*n* = 88, 16.8%) (Table 1).

### 3.2. Psychosocial Characteristics by Occupation 

A comparison of psychosocial characteristics by occupation revealed that PS (18.93 ± 5.48) had a significantly higher score on the COVID-19 anxiety scale than 119 REMS (*F* = 3.396, *p* < 0.001). With regard to anxiety symptoms, PHC (6.03 ± 3.75) scored significantly higher than 119 REMS (2.95 ± 2.75) and police officers (3.60 ± 2.56), and PS (5.31 ± 3.23) scored significantly higher than 119 REMS and police officers (*F* = 18.440, *p* < 0.001). For depressive symptoms, PHC (8.17 ± 3.72) scored significantly higher than 119 REMS (4.22 ± 3.40) and police officers (5.35 ± 3.22), and PS (6.91 ± 3.72) scored significantly higher than 119 REMS and police officers (*F* = 17.440, *p* < 0.001). As for somatic symptoms, PHC (6.71 ± 4.88) scored significantly higher than 119 REMS (3.17 ± 3.30) and police officers (4.38 ± 3.44), and PS (6.65 ± 4.71) also scored significantly higher than 119 REMS and police officers (*F* = 18.796, *p* < 0.001). Regarding quality of life, 119 REMS (180.64 ± 26.20) scored significantly higher than PHC (165.76 ± 23.73) and PS (163.90 ± 23.60), while police officers (176.87 ± 23.17) scored significantly higher than PS (163.90 ± 23.60) (*F* = 12.373, *p* < 0.001). In terms of resilience, 119 REMS (69.80 ± 16.24) scored significantly higher than PHC (60.81 ± 16.66) and PS (60.36 ± 14.98), and police officers (66.94 ± 13.75) scored significantly higher than PS (60.36 ± 14.98) (*F* = 9.334, *p* < 0.001). However, there were no significant differences between occupations with regard to sleep (Table 2).

### 3.3. Quality of Life Characteristics by Occupation 

The groups showed differences in the quality of life and all of its domains, including general well-being (*F* = 15.971, *p* < 0.001), physical health (*F* = 11.754, *p* < 0.001), psychological health (*F* = 14.490, *p* < 0.001), social relationships (*F* = 9.057, *p* < 0.001), and environment (*F* = 9.012, *p* < 0.001). In the post hoc test, PHC (6.37 ± 1.43) scored significantly lower than 119 REMS (7.61 ± 1.40) and police officers (7.10 ± 1.26) with regard to general well-being, and PS (6.57 ± 1.37) also scored significantly lower than 119 REMS and police officers. In terms of the physical domain, PHC (23.49 ± 3.85) scored significantly lower than 119 REMS (25.53 ± 3.41), and PS (22.99 ± 4.17) scored significantly lower than 119 REMS and police officers (25.17 ± 3.77). With regard to the psychological domain, PHC (20.05 ± 3.40) also scored significantly lower than 119 REMS (22.98 ± 3.30) and police officers (20.00 ± 3.17), and PS (20.57 ± 3.35) scored significantly lower than 119 REMS and police officers. As for the social domain, PS (10.01 ± 1.66) scored significantly lower than 119 REMS (11.10 ± 1.97) and police officers (10.76 ± 1.88). Lastly, for the environment domain, PS (28.12 ± 4.20) scored significantly lower than 119 REMS (30.54 ± 4.24) and police officers (30.00 ± 4.34) (Table 3).

### 3.4. Correlation between Quality of Life and Psychological Variables

Quality of life was found to correlate positively with resilience (r = 0.942, *p* < 0.01) and negatively with COVID-19 anxiety (r = −0.154, *p* < 0.01), depressive symptoms (r = −0.593, *p* < 0.01), anxiety (r = −0.545, *p* < 0.01), somatic symptoms (r = −0.426, *p* < 0.01), and insomnia (r = −0.160, *p* < 0.01) (Table 4).

### 3.5. Factors Affecting the Quality of Life

Stepwise regression analysis was performed to identify the variables that explain the quality of life. This study found that resilience (β = 0.897, *p* < 0.01) was the most significant explanatory variable, and, together with insomnia (β = 0.154, *p* < 0.01), depression (β = −0.152, *p* < 0.01), and COVID-19 anxiety (β = −0.057, *p* < 0.01), accounted for 91.8% of the explanatory variance with regard to quality of life (Table 5).

## 4. Discussion

The impact of the COVID-19 pandemic has not been limited to physical problems. It has also affected people’s mental health. In particular, the measures being taken to prevent the spread of the virus, such as social distancing, the use of protective equipment, quarantine, and isolation are causing psychological symptoms such as depression, anxiety, and insomnia in socially vulnerable groups. Disaster service workers, in particular, experience extreme stress on the frontlines, yet there has been a lack of reports on their mental health to date. Disaster service workers have a higher trauma exposure than ordinary people and can suffer secondary trauma in the process of dealing with victims even if they are not directly exposed, so the morbidity of psychiatric disorders such as depression, anxiety, and alcohol use disorder is high [32]. Therefore, this study aimed to investigate the factors affecting the quality of life and psychosocial characteristics of disaster service workers who are employed in COVID-19-related work. 

In this study’s findings, public servants of city hall (PS) had higher levels of anxiety regarding COVID-19 than those working in 119 rescue and emergency medical services (119 REMS). In addition, PS and those working in public health care centers (PHC) had higher levels of depression, anxiety, and somatic symptoms, as well as a lower quality of life when compared to 119 REMS and police officers. In terms of resilience, 119 REMS scored higher than PHC, and 119 REMS and police officers scored higher than PS.

According to previous studies, the longer the time spent in contact with patients in a COVID-19-related situation, the greater the psychological symptoms, such as depression and anxiety [33]. In particular, white-collar workers perform significant mentally stressful activities but little physical activity, so it is important that attention is paid to both their physical and mental health [34]. In the case of PHC and PS, who engage in white-collar work, the following factors are thought to worsen psychopathological symptoms such as anxiety and depression: long-term COVID-19-related general administrative work, job stress, difficulties with filed complaints, emotional labor, stress caused by verbal abuse, and coping with the lengthening pandemic. 

It is difficult to clearly determine the cause of somatization or explain it, anatomically and psychologically [35]. In most cases, patients complain of physical symptoms that are complex and diverse rather than simple, and such symptoms have an impact on their social roles and occupational functioning. Depression, which is known to be a representative mediating factor [36], plays a role in amplifying physical sensations [37]. Therefore, in the case of PHC and PS, somatization scores may be high due to the influence of high levels of depression and anxiety. On a secondary note, the increased levels of anxiety resulting from the risk of exposure and infection, as well as the difficulties caused by intensive work [38], may have been expressed through somatization. 

This study found no differences between groups with regard to sleep. As the COVID-19 pandemic lengthened, PHC and PS had to undergo changes in their work environments, such as having to work overtime and in shifts due to surges in workloads and performing various additional tasks [39]. They took on COVID-19-related work during the day and managed their regular workload during the night, experiencing sleep disorders as a result of working hours that did not match their circadian rhythms [40]. On the contrary, in the case of the 119 REMS and police officer groups, many already had sleep problems [41] because they often worked two- or three-shift work schedules, which may be why there were no differences in the groups with regard to sleep [42]. The reduced resilience in PHC and PS could be attributed to the high levels of anxiety and depression affecting their psychopathology, insomnia caused by sleep disorders, and increased job stress due to COVID-19 [43].

An individual’s quality of life is related to their physical health, psychological state, degree of independence, social relationships, personal beliefs, and relationship with their environment [44]. In this study, it was determined that PHC and PS had a low score for all the quality of life domains, which may be due to somatization that resulted from the various psychological factors identified above, changes in work environments as a result of unexpected disaster situations, increased workloads, and intensive work, as well as the stress of on-site responses [45]. In addition, it appears that PHC and PS experienced an overall decreased quality of life as a result of isolating themselves rather than engaging in social activities, which was due to the anxiety that their family and friends might become infected because of their exposure to infection [46].

Quality of life was found to be negatively correlated with insomnia, depression, and COVID-19 anxiety while being positively correlated with resilience. Resilience refers to an individual’s ability to cope with and overcome adversity and stressful situations [47]. According to previous studies, resilience acts as a protective factor by affecting a person’s coping strategies in stressful situations that lead to anxiety and depression [48]. Therefore, promoting resilience is considered to be an important task to improve quality of life. Resilience is a dynamic concept affected on a multidimensional level by factors such as individual characteristics, family, interpersonal relationships, and external environments [49], and it can be strengthened through training and learning [50]. Therefore, active interventions are needed to improve the resilience of disaster service workers by ensuring adequate sleep, a positive emotional state [51], mindfulness, and appropriate stress management techniques [52]. 

The limitations of this study are as follows: First, as the subjects of this study are disaster service workers (PHC, PS, 119 REMS, and police officers) who engage in COVID-19-related work, the generalizability of the findings is limited. Second, the proportion of white-collar workers among the study subjects is high. Third, as this is a cross-sectional study, its causal reasoning is limited. Fourth, the differences in subjects according to whether or not they had been vaccinated were not investigated. 

Regardless of the limitations, this study is significant because there are studies that evaluate mental health in patients in the COVID-19 situation, but there are no reports of mental health evaluation in disaster service workers. Thus, this study lays the theoretical groundwork for improving the quality of life and mental health of disaster service workers by investigating their mental health in the context of the COVID-19 pandemic. 

## 5. Conclusions

Healthcare workers and disaster service workers have been reported to be vulnerable to mental health problems during outbreaks of infectious diseases such as the COVID-19 pandemic. Regarding quality of life, 119 REMS scored significantly higher than PHC and PS, while police officers scored significantly higher than PS. Resilience was the most significant explanatory variable, and, together with insomnia, depression, and COVID-19 anxiety, accounted for 91.8% of the explanatory variance with regard to quality of life. Quality of life was found to be negatively correlated with insomnia, depression, and COVID-19 anxiety, while being positively correlated with resilience. Therefore, active interventions are needed to improve the resilience of disaster service workers.

## Figures and Tables

**Table 1 jcm-11-01600-t001:** Demographic and clinical characteristics of participants.

		Variables
Sex	Male	259 (49.2%)
	Female	267 (50.8%)
Age	20s	134 (25.5%)
	30s	159 (30.2%)
	40s	100 (19.0%)
	50s	128 (24.3%)
	60s	5 (1%)
Occupation	Public Healthcare Center	66 (12.5%)
	Public servant at city hall	250 (47.5%)
	119 REMS	122 (23.2%)
	Police officer	88 (16.8%)
Marital status	Single	210 (39.9%)
	Married	303 (57.6%)
	Divorced	9 (1.7%)
	Seperated	4 (0.2%)
Education (years)	Low (<12)	65 (12.4%)
	Middle (12–16)	73 (13.9%)
	High (>16)	388 (73.89%)
Chronic disease	Yes	63 (12.0%)
	No	463 (88.0%)
Respiratory disease	Yes	37 (7.0%)
	No	489 (93.0%)

119 REMS: 119 Rescue and Emergency Medical Service.

**Table 2 jcm-11-01600-t002:** Difference of KF-COVID-19S, HADS, PHQ-15, ISI, and CD-RISC among the groups (*n* = 526).

Variables		Public Healthcare Center (*n* = 66)	Public Servant at City Hall (*n* = 250)	119 REMS (*n* = 122)	Police Officer (*n* = 88)	*F*	Post-Hoc (Bonferroni)
KF-COVID-19S (M ± SD)		18.08 ± 4.96	18.93 ± 5.48	17.03 ± 5.10	18.05 ± 5.68	3.396 **	3 < 2
HADS (M ± SD)	Anxiety	6.03 ± 3.75	5.31 ± 3.23	2.92 ± 2.75	3.60 ± 2.56	18.440 ***	3 < 1, 4 < 1, 3 < 2, 4 < 2
	Depression	8.17 ± 3.72	6.91 ± 3.72	4.22 ± 3.40	5.35 ± 3.22	17.440 ***	3 < 1, 4 < 1, 3 < 2, 4 < 2
PHQ-15 (M ± SD)		6.71 ± 4.88	6.65 ± 4.71	3.17 ± 3.30	4.38 ± 3.44	18.796 ***	3 < 1, 4 < 1, 3 < 2, 4 < 2
ISI (M ± SD)		7.63 ± 5.09	7.34 ± 5.10	5.88 ± 4.26	7.65 ± 5.89	2.782 *	ns
CD-RISC (M ± SD)		60.81 ± 16.66	60.36 ± 14.98	69.80 ± 16.24	66.94 ± 13.75	9.334 ***	1 < 3, 2 < 3, 2 < 4

* *p* < 0.05, ** *p* < 0.01, *** *p* < 0.001, *n*: number, M: mean, SD: standard deviation, 119 REMS: 119 Rescue and Emergency Medical Service, KF-COVID-19S: The Korean version of the Fear of COVID-19 Scale, HADS: Hospital Anxiety Depression Scale, ISI: Insomnia severity scale, PHQ-15: Patient Health Questionnaire-15, CD-RISC: Connor-Davidson Resilience Scale, 1: Public Healthcare Center, 2: Public servant at city hall, 3: 119 REMS, 4: Police officer, ns: non-specific.

**Table 3 jcm-11-01600-t003:** Comparison of WHOQOL-BREF among the groups (*n* = 526).

Variables	Public Healthcare Center (*n* = 66)	Public Servant at City Hall (*n* = 250)	119 REMS (*n* = 122)	Police Officer (*n* = 88)	*F*	Post-Hoc (Bonferroni)
Overall	6.37 ± 1.43	6.57 ± 1.37	7.61 ± 1.40	7.10 ± 1.26	15.971 ***	1 < 3, 1 < 4, 2 < 3, 2 < 4
Physical	23.49 ± 3.85	22.99 ± 4.17	25.53 ± 3.41	25.17 ± 3.77	11.754 ***	1 < 3, 2 < 3, 2 < 4
Psychological	20.05 ± 3.40	20.57 ± 3.35	22.98 ± 3.30	20.00 ± 3.17	14.490 ***	1 < 3, 1 < 4, 2 < 3, 2 < 4
Social	10.75 ± 1.73	10.01 ± 1.66	11.10 ± 1.97	10.76 ± 1.88	9.057 ***	2 < 3, 2 < 4
Environmental	28.90 ± 0.4.42	28.12 ± 4.20	30.54 ± 4.24	30.00 ± 4.34	9.012 ***	2 < 3, 2 < 4
Sum of WHOQOL-BREF	165.76 ± 23.73	163.90 ± 23.60	180.64 ± 26.20	176.87 ± 23.17	12.373 ***	1 < 3, 2 < 3, 2 < 4

*** *p* < 0.001, *n*: number, M: mean, SD: standard deviation, 119 REMS: 119 Rescue and Emergency Medical Service, WHOQOL-BREF: Korean Version of World Health Organization Quality of Life Assessment Instrument Brief Form, 1: Public Healthcare Center, 2: Public servant at city hall, 3: 119 REMS, 4: Police officer.

**Table 4 jcm-11-01600-t004:** Bivariate associations between quality of life and psychological variables (*n* = 526).

Variables	KF-COVID-19S	HADS (Anxiety)	HADS (Depression)	PHQ-15	ISI	CD-RISC	WHOQOL-BREF
KF-COVID-19S	1						
HADS (Anxiety)	0.351 **	1					
HADS (Depression)	0.236 **	0.729 **	1				
PHQ-15	0.251 **	0.655 **	0.599 **	1			
ISI	−0.151 **	0.413 **	−0.368 **	−0.504 **	1		
CD-RISC	−0.094 *	−0.516 **	−0.540 **	−0.433 **	0.278 **	1	
WHOQOL-BREF	−0.154 **	−0.545 **	−0.593 **	−0.426 **	−0.160 **	0.942 **	1

* *p* < 0.05, ** *p* < 0.01. KF-COVID-19S: The Korean version of the Fear of COVID-19 Scale, WHOQOL-BREF: World Health Organization quality of life assessment instrument brief form, HADS: Hospital Anxiety Depression Scale, PHQ-15: Patient health questionnaire-15, ISI: Insomnia severity scale, CD-RISC: Connor–Davidson Resilience Scale.

**Table 5 jcm-11-01600-t005:** Stepwise regression analysis of quality of life among the subjects (*n* = 526).

	Standardized β	*t*	*p*	Adjusted R²	*F*	*p*
CD-RISC	0.897	60.102	<0.01	0.918	1474.197	<0.001
ISI	0.154	11.398	<0.01			
HADS (Depression)	−0.152	−9.679	<0.01			
KF-COVID-19S	−0.057	−4.444	<0.01			

CD-RISC: Connor–Davidson resilience scale, ISI: Insomnia severity scale, HADS: Hospital Anxiety Depression Scale, KF-COVID-19S: The Korean version of the Fear of COVID-19 Scale.

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
