# Peer review of "Mental Health and Quality of Life for Disaster Service Workers in a Province under COVID-19"

_jcm, 2022, doi:10.3390/jcm11061600_

Round 1

Reviewer 1 Report

Badanie piÄ™ciu wybranych grup pracowników sÅ‚użby zdrowia i pomocy w przypadku katastrof najbardziej zagrożonych pandemiÄ… COVID-19 byÅ‚o bardzo przemyÅ›lanym badaniem. Pięć kwestionariuszy zostaÅ‚o przebadanych przez 526 osób, w tym zarówno nowych, takich jak KF-COVID-19S, ale także wykorzystywanych do tej pory w innych badaniach dotyczÄ…cych zdrowia psychicznego, lÄ™ku szpitalnego, depresji i bezsennoÅ›ci, a także skali odpornoÅ›ci Connor-Davidson i WHOQOL-BREF. Bardzo dobre, dogłębne statystyki, w tym stopniowana analiza regresji w celu zidentyfikowania czynników wpÅ‚ywajÄ…cych na jakość życia. Odkrycia potwierdzajÄ… inne podobne badania, które mówiÄ…, że jakość życia jest negatywnie skorelowana z bezsennoÅ›ciÄ…, depresjÄ… i lÄ™kiem COVID-19, podczas gdy pozytywnie skorelowana z odpornoÅ›ciÄ…. Zalecenia na przyszÅ‚ość zostaÅ‚y dobrze okreÅ›lone. 49 referencji obejmowaÅ‚o zarówno książki, jak i czasopisma z lat 2002-2022. Bardzo interesujÄ…cy obszar badaÅ„ i wskazane byÅ‚oby zaplanowanie wielooÅ›rodkowego badania. Gratulacje

Author Response

Thank you for your valuable comments.

Reviewer 2 Report

Current study designed to investigate the psychosocial characteristics of disaster service workers in charge of COVID-19 related work in addition to identify the factors affecting their quality of life. I like to give the following comments.

  1. This study from June 2020 to June 2021 that needs a rationale to introduce.
  2. Patient Health Questionnaire-15 needs the reliability from reference(s).
  3. Hospital Anxiety and Depression Scale (HADS) was defined by who and how?
  4. Same as Insomnia Severity Index (ISI), questionnaire for Quality of Life, and Connor-Davidson Resilience Scale (CD-122 RISC),
  5. Mental health in disaster service workers during regular stage without COVID-19 pandemic shall be cited to compare in discussion.
  6. It is interesting to know the variations between vaccination and not in the participants.
  7. In the discussion, insomnia caused by sleep disorders, and increased job stress due to COVID-19 that needs reference(s) to support.
  8. Novelty of the current study was not indicated in the conclusion.

Author Response

Dear reviewer

1. This study from June 2020 to June 2021 that needs a rationale to introduce.

-->It is marked in red in the text.

A survey was conducted of 526 disaster service workers in charge of COVID-19-related work who understood this study’s purpose and agreed to participate from June 2020 to June 2021, when the mental health related to COVID-19 began to emerge.

2. Patient Health Questionnaire-15 needs the reliability from reference(s).

--> It is marked in red in the text.

.Han et al. [22] translated the questionnaire into Korean and verified its validity and reliability.

3. Hospital Anxiety and Depression Scale (HADS) was defined by who and how?

  • It is marked in red in the text. (added references)

4. Same as Insomnia Severity Index (ISI), questionnaire for Quality of Life, and Connor-Davidson Resilience Scale (CD-122 RISC),

  • It is marked in red in the text. (added references)

5. Mental health in disaster service workers during regular stage without COVID-19 pandemic shall be cited to compare in discussion.

  • Disaster service workers have higher trauma exposure than ordinary people and can suffer secondary trauma in the process of dealing with victims even if they are not directly exposed, so the mobidity of psychiatric disorders such as depression, anxiety, and alcohol use disorder is high. (Lee, S.H.; Kim, S.J., Sim, M.Y., Yoo, S.Y., Won, S.D., Lee, B.C. Mental Health of diaster workers. J Korean Neuropsychiatr Assoc 2015;54(2):135-141.)
  •  
  •  

6. It is interesting to know the variations between vaccination and not in the participants.

--> In this study, the differences in subjects according to whether or not they are vaccinated were not investigated. (Added it to limitation)

7. In the discussion, insomnia caused by sleep disorders, and increased job stress due to COVID-19 that needs reference(s) to support.

  • It is marked in red in the text. (added references)

[41] LI, Y. Cehn, B., Hong, Z., Sun, Q., Dai, Y., Basta, M., Qul, Q. Insomnia symptoms during the early and late stages of the COVID-19 pandemic in china; a systemic review and meta-analysis. Sleep Med 2021, doi: 10.1016/j.sleep.2021.09.014

[42] Zou, X., Liu, S., Li, J., Chen, W., Ye, J., Yang, Y., Ling, L. Factors associated with healthcare worker’s insomnia symptoms and fatigue in the fight against COVID-19, and the role of organizational support. Front.Psychiatry 12:652717.

8, Novelty of the current study was not indicated in the conclusion.

  • It is marked in red in the text.

Regardless of the limitations, this study is significant because there are studies that evaluate mental health in patients in the COVID1-9 situation, but there are no reports of mental health evaluationin dasaster service workers. So, it laid the theoretical groundwork for improving the quality of life and mental health of disaster service workers by investigating their mental health in the context of the COVID-19 pandemic.

Thanky you for your valuable comment.
